# Reading Your Heart: Learning ECG Words and Sentences via Pre-training ECG Language Model

Jiarui Jin[*]
South China Normal University
Foshan, China

Haoyu Wang[*]
South China Normal University
Foshan, China

Jun Li
Jilin University
Changchun, China

Sichao Huang
South China Normal University
Foshan, China

Jiahui Pan[†]
South China Normal University
Foshan, China

Shenda Hong[†]
Peking University
Beijing, China

## ABSTRACT

Electrocardiograms (ECGs) are essential for the clinical diagnosis of arrhythmias and other heart diseases, but deep learning methods based on ECGs often face limitations due to the need for high-quality annotations. Although previous ECG self-supervised learning (eSSL) methods have made significant progress, they typically treat ECG signals as general time-series data, using fixed steps and window sizes, which often ignore the heartbeat and rhythmic characteristics and potential semantic relationships in ECG signals. In this work, we introduce a novel perspective on ECG signals, treating heartbeats as words and rhythms as sentences. Based on this perspective, we propose HeartLang, a novel self-supervised learning framework for ECG language processing. Within this framework, we construct an ECG vocabulary and pre-train the model using masked prediction on ECG sentences to learn both heartbeat-level and rhythm-level representations, uncovering the latent semantic relationships in ECG signals. We also developed three parameter scales for HeartLang, namely, HeartLang-Small, HeartLang-Base, and HeartLang-Large, and conducted pre-training and downstream task testing on the standard benchmark dataset PTB-XL.The experimental results demonstrate that our method exhibits superior performance compared to other eSSL methods.

## CCS CONCEPTS

• **Computing methodologies** → *Knowledge representation and reasoning.*

## KEYWORDS

Electrocardiogram (ECG), ECG language processing, self-supervised learning, transformer.

**ACM Reference Format:**
Jiarui Jin, Haoyu Wang, Jun Li, Sichao Huang, Jiahui Pan, and Shenda Hong. 2024. Reading Your Heart: Learning ECG Words and Sentences via Pre-training ECG Language Model. In *Proceedings of (KDD-AIDSH '24)*. ACM, New York, NY, USA, 12 pages. https://doi.org/10.1145/nnnnnnn.nnnnnnn

[*]Both authors contributed equally to this research. This work was done when Jiarui Jin was an intern at National Institute of Health Data Science, Peking University
[†]Corresponding authors:
Shenda Hong (hongshenda@pku.edu.cn)
Jiahui Pan (panjiahui@m.scnu.edu.cn)

*KDD-AIDSH '24, August 25–29, 2024, Barcelona, Spain*
2024. ACM ISBN 978-x-xxxx-xxxx-x/YY/MM...$15.00
https://doi.org/10.1145/nnnnnnn.nnnnnnn

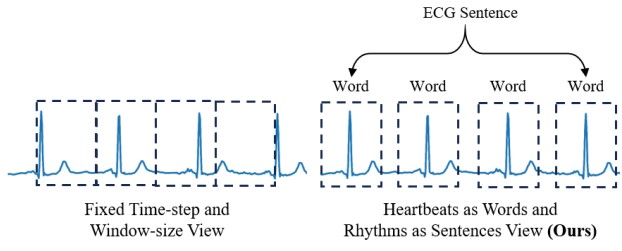

**Figure 1: Two perspectives on ECG signals.**

## 1 INTRODUCTION

Electrocardiogram (ECG) is a common type of clinical data used to monitor cardiac activity, and is frequently employed in diagnosing cardiac diseases or conditions impairing myocardial function[12]. A primary limitation of using supervised deep learning methods for ECG signal analysis is their dependency on large-scale, expert-reviewed, annotated high-quality data. Moreover, even with sufficient data, these methods are often designed to address specific tasks, which curtails the generalization ability of the model. To overcome these challenges, ECG self-supervised learning (eSSL) has demonstrated significant efficacy by training on vast amounts of unlabeled cardiac recordings to derive generic ECG signal representations, which are then fine-tuned for specific downstream tasks[16][5][9].

Current eSSL methods can be primarily classified into two categories: contrastive-based methods and reconstruction-based methods. The core principle of contrastive-based methods involves creating positive and negative sample pairs, with the objective of maximizing the similarity of positive pairs and minimizing the similarity of negative pairs in the feature space. Reconstruction-based methods focus on training a model to reconstruct the original input from partial or transformed data, thereby learning effective data representations. However, almost all methods treat ECG signals as ordinary time series data. These methods have two significant drawbacks:

**Ignoring Heartbeat and Rhythmic Characteristics of ECG.** In ECG diagnostics, heartbeat characteristics are essential. For example, myocardial infarction is diagnosed by observing ST segment elevation[15]. Likewise, cardiac rhythm characteristics are critical, as arrhythmias are identified based on the overall cardiac rhythm[1]. However, existing eSSL methods typically employ fixed-size and

fixed-step time windows to segment the data. This method treats ECG signals as ordinary time series signals, thus neglecting the unique heartbeat and rhythmic features inherent in ECG signals.

**Ignoring Latent Semantic Relationships of ECG.** Due to significant differences in heart rate and other factors between different subjects, and even among different samples from the same subject, using fixed-size and fixed-step time windows to segment data leads to substantial discrepancies among samples. The learning of latent semantic representations depends on the similarity between samples, but the variability introduced by fixed-size and fixed-step time windows disrupts this similarity. Different samples are not processed within a unified feature space, resulting in inconsistent feature representations. This lack of uniformity hinders the model's ability to find effective similarities when learning latent semantic representations, leading to suboptimal learning outcomes.

To address these challenges, we propose a novel self-supervised learning framework for ECG language processing (ELP) [13] named **HeartLang**. A significant distinction between ECG and other physiological signals is the clarity of heart rate over time, where most heartbeats can be distinctly recognized and differentiated. The core idea of this framework is to treat **heartbeats as words and rhythms as sentences**. First, we segment the ECG signal using QRS complexes to obtain the original heartbeat intervals. Then, we concatenate the heartbeat intervals from different leads to form ECG sentences. To overcome inter-subject variability, we do not directly treat the original heartbeat intervals as ECG words. Instead, we construct a large ECG vocabulary, quantifying heartbeats from different subjects into discrete ECG sentence embeddings, and use these embeddings to reconstruct the original heartbeat intervals to learn representations of the same heartbeat across different subjects. ECG vocabulary provides a compact, general, and meaningful representation of different heartbeat intervals. To enable the framework to learn representations of the overall ECG rhythm, we mask parts of the ECG sentences and predict the missing ECG words during large-scale pre-training. Through these approaches, our method can learn both heartbeat-level and rhythm-level representations of ECG signals without labels, and extract latent semantic representations in ECG sentences using ECG vocabulary and masked prediction pre-training. The main contributions of this work are summarized in below:

- We propose HeartLang, a novel self-supervised learning framework for ECG language processing, designed to learn information-rich representations at different levels from unlabeled ECG signals.
- We introduce an innovative perspective on ECG signals, viewing heartbeats as words and rhythms as sentences, and extract latent semantic representations in ECG signals using self-supervised learning.
- We construct a large-scale ECG vocabulary to enable the framework to learn heartbeat-level features via reconstruction, while using masked prediction during large-scale pre-training on ECG sentences to learn rhythm-level features, thereby allowing the framework to extract generalized representations at different levels.

- We designed ECGFormer, a novel Transformer-based backbone network for ECG signals. It leverages the spatiotemporal information in ECG signals to enhance representation extraction and optimize feature extraction for long ECG sentences.

## 2 METHODS

In this section, we provide a detailed explanation of the specific structure of the HeartLang framework. We first define multi-lead ECG data as $X \in \mathbb{R}^{C \times T}$, where $C$ represents the number of ECG leads (electrodes) and $T$ represents the total timestamps. The configuration of ECG leads follows the standard 12-lead ECG setup. The overview of the framework is shown in Figure 2. The use of the framework can be divided into three steps. First, constructing the ECG vocabulary is achieved through the steps in the Section 2.3. Second, masked pre-training of the framework is performed as described in the Section 2.4. Finally, fine-tuning for downstream tasks is carried out, which can be divided into linear probing, where only the classification head is trained in the ECGFormer, and full parameter fine-tuning, where all parameters of the ECGFormer are trained.

### 2.1 QRSTokenizer

A key concept of our method is to treat heartbeats as words, thus making the segmentation of the original ECG signal into heartbeat sequence essential. We introduce QRSTokenizer, a tokenizer that segments ECG signals into heartbeat sequences based on QRS waveforms. Initially, we bandpass filter the I-lead signal between 5 and 20 Hz, apply moving wave integration (MWI) with a Ricker wavelet to the filtered signal, and save the squared integration signal. We then traverse the local maxima of the MWI signal; each local maximum that meets the criteria of being post-refractory period and exceeding the QRS detection threshold is classified as a QRS complex. Following detection, we obtain the indices of the detected QRS complexes $Q = \{q_i | i = 1, \ldots, N\}$, where $N$ denotes the number of detected QRS indices per sample, which varies between samples.

Assuming the time window size is $t$. For each lead, we center each index in $Q$, using the midpoint between every two indices as the interval boundaries, and independently segment the QRS complex intervals for each lead. If the segmented region is smaller than $t$, we pad it with zeros to match the required size. After segmentation, we concatenate the heartbeat intervals of the 12 leads in sequence, forming the overall ECG sentence $x \in \mathbb{R}^{l \times t}$, where $l$ represents the sequence length and $t$ denotes the time window size. Given the variability in heart rates across samples, the resulting sequence lengths are inconsistent. Similar to natural language processing, we set $l$ to the maximum length of the ECG sentence. If the length of the ECG sentence is less than $l$, it will be padded to $l$ through the zero interval; if the length of the ECG sentence exceeds $l$, the interval length will be truncated to $l$.

### 2.2 ECGFormer Backbone Network

To construct an ECG vocabulary and predict these vocabularies during pre-training, we designed a backbone network capable of extracting and reconstructing the semantic representations contained in ECG sentences. This backbone network is employed in

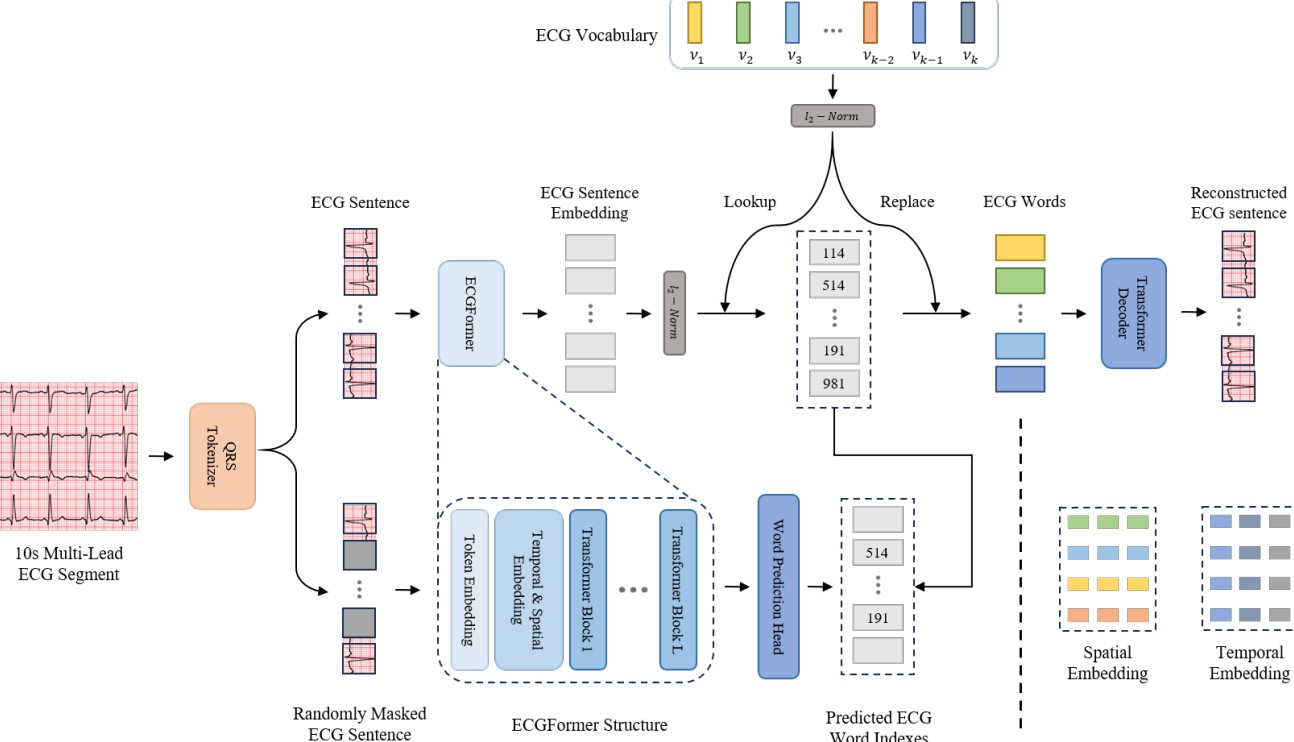

Figure 2: Framework of HeartLang.

various components of the HeartLang framework, including Vector-Quantized Heartbeat Reconstruction, Pre-training, and Supervised Fine-Tuning, and is named ECGFormer. The ECGFormer inputs a sample $X \in \mathbb{R}^{l \times t}$, where $l$ represents the maximum length of the ECG sentence, and $t$ has different meanings in different parts of the framework. For example, in heartbeat reconstruction, $t$ actually represents the dimension of the ECG words, but in vector quantization, it figuratively represents the size of the time window of the input sample. The main components of ECGFormer include ECG Sentence Projection, temporal and spatial embedding, Rotational Position Encoding (ROPE), and multi-layer Transformer Encoder.

2.2.1 *Token Embedding.* ECG signals have high temporal resolution, and the QRS complexes forming ECG sentences contain rich temporal features. By mapping these QRS complexes into a new feature space, we can more effectively extract and encode these features. We use a 1-D convolutional layer-based mapper to convert heartbeats in ECG sentences into tokens. After mapping, each heartbeat interval in the ECG sentence can be represented as $e = \{e_{i,j} \in \mathbb{R}^D \mid i = 0, 1, \ldots, C, j = 0, 1, \ldots, T\}$, where $D$ is the dimension of the elevated feature space. Here, $e_{0,0}$ represents the padded heartbeat interval, which will be explained in detail in section 2.2.2. The mapped ECG sentence is represented as: $X \in \mathbb{R}^{l \times D}$.

2.2.2 *Temporal & Spatial Embedding.* In our designed ECG sentences, heartbeat intervals have two inherent characteristics: different leads and varied time positions. For example, in one sample, the centers of the first two QRS complexes are within the first second,

while in other samples, only one QRS complex center falls within this interval.

To enable the ECGFormer to better capture the temporal and spatial information of heartbeat intervals in ECG sentences and to understand the differences between various heartbeat intervals, we initialized a temporal embedding list $TL = \{t_0, t_1, t_2, \ldots, t_T\}$ and a spatial embedding list $SL = \{s_0, s_1, s_2, \ldots, s_C\}$, both of which are $D$-dimensional and learnable during the training process.

As described in section 2.1, each heartbeat interval can be identified by its lead and its time window. For the time window of each heartbeat interval, we divided the original data into 10-second segments. The specific time window of each heartbeat interval depends on the peak of the QRS complex within that interval. However, due to zero-padding applied to align the ECG sentence to the maximum length, we need additional identification. In this paper, we set both the lead and time window of zero-padded intervals to 0. This approach simplifies and extends the process of adding temporal and spatial embeddings, as it does not consider the number of leads or time windows.

Temporal embedding $t_0$ and spatial embedding $s_0$ are used to distinguish whether data within a heartbeat interval is zero-padded; only when an interval is zero-padded do we add the $t_0$ temporal embedding and the $s_0$ spatial embedding to it. In other cases, we add temporal and spatial embeddings based on the lead and the time window the heartbeat interval belongs to. Thus, any heartbeat interval $e_{i,j}$ from the mapper, after adding temporal and spatial

embeddings, is represented as:

$$e = \{e_{i,j} + s_i + t_j | i = 0, 1, \ldots, C, j = 0, 1, \ldots, T\}$$

Here, the temporal and spatial embeddings serve as absolute position encodings.

*2.2.3 Rotary Position Embedding (ROPE).* In addition to using temporal and spatial embeddings to capture the absolute positional relationships within ECG sentences, we also introduced Rotary Position Embedding (ROPE) to enhance the ECGFormer's understanding of relative positional relationships. This dual encoding approach makes the ECGFormer more flexible and comprehensive in understanding ECG sentences. Assume the input vector is $X \in \mathbb{R}^{l \times D}$, with dimension $D$ and sequence length $l$. ROPE incorporates positional information directly into the features of heartbeat intervals by applying a rotational transformation to the position encoding. The feature representation after applying ROPE, denoted as ROPE($X$), is calculated as follows: First, define an angle matrix $\Theta$, where each element $\theta_{i,j}$ represents the angle for position $i$ and dimension $j$:

$$\theta_{i,j} = \frac{i}{10000^{\frac{2j}{D}}}$$

for $i = 0, 1, 2, \ldots, l-1$ and $j = 0, 1, 2, \ldots, \frac{D}{2} - 1$. Construct a rotation matrix $R(\theta)$ to apply the rotational transformation to the input vectors:

$$R(\theta) = \begin{bmatrix} \cos(\theta) & -\sin(\theta) \\ \sin(\theta) & \cos(\theta) \end{bmatrix}$$

Split the input vector $X$ into two sub-vectors $X_{\text{even}}$ and $X_{\text{odd}}$:

$$X_{\text{even}} = [X_{i,2j}] \quad \forall i \in [0, l), j \in [0, \frac{D}{2})$$

$$X_{\text{odd}} = [X_{i,2j+1}] \quad \forall i \in [0, l), j \in [0, \frac{D}{2})$$

Finally, for each position $i$ and dimension $j$, apply the rotation matrix $R(\theta_{i,j})$ to transform the vectors:

$$\text{ROPE}(X)_{i,2j} = X_{i,2j} \cos(\theta_{i,j}) + X_{i,2j+1} \sin(\theta_{i,j})$$

$$\text{ROPE}(X)_{i,2j+1} = -X_{i,2j} \sin(\theta_{i,j}) + X_{i,2j+1} \cos(\theta_{i,j})$$

The new feature representation after rotation, denoted as ROPE($X$), is:

$$\text{ROPE}(X) = [\text{ROPE}(X)_{i,2j}, \text{ROPE}(X)_{i,2j+1}] \quad \forall i \in [0, l), j \in [0, \frac{D}{2})$$

The final feature representation encodes each position $i$ and dimension index $j$ in the entire input sequence $X$, meaning that each element ROPE($X$)$_{i,k}$ contains a combination of the original feature $X$ and positional information, where $k = 0, 1, \ldots, D-1$.

*2.2.4 Transformer Encoder.* Finally, the heartbeat interval feature representation obtained by adding ROPE, denoted as ROPE($X$), is directly used to compute attention scores. This captures the complex temporal and spatial relationships between different heartbeat intervals in the ECG sentence, thus enhancing the ECGFormer's regression and classification capabilities. The representation of the ECG sentence with added temporal and spatial embeddings is $X \in \mathbb{R}^{l \times D}$, on which the rotational position encoding is added and self-attention is computed. The computation process is as follows:

$$Q = \text{ROPE}(XW^Q), \quad K = \text{ROPE}(XW^K), \quad V = XW^V$$

$$\text{Attention}(Q, K, V) = \text{softmax}\left(\frac{QK^{\mathsf{T}}}{\sqrt{d_k}}\right) V$$

## 2.3 Vector-Quantized Heartbeat Reconstruction

Before pre-training HeartLang with masking and prediction, we need to tokenize the original heartbeat intervals in the ECG sentence as ECG words. We introduce vector-quantized heartbeat reconstruction, which uses discrete ECG word vectors to reconstruct the original heartbeat intervals, thereby training the ECG vocabulary, as shown in the upper part of Figure 2. During training, the representation of these discrete ECG words is optimized by minimizing the reconstruction error. Similar heartbeat intervals of different subjects are mapped to the same discrete vector, thereby overcoming the physiological differences between subjects and providing a compact and universal ECG word for subsequent pre-training. The core components of this method are the ECG vocabulary and transformer decoder. The ECG vocabulary maps each original heartbeat interval in the ECG sentence to a discrete ECG word, while the transformer decoder decodes the ECG word back to the original heartbeat interval. This concept is inspired by VQ-NSP[8], which encodes EEG signals into discrete latent representations and decodes them.

*2.3.1 Vector Quantization.* We first define an ECG vocabulary $\mathcal{V} = \{v_i | i = 1, \ldots, K\} \in \mathbb{R}^{K \times D}$, where $K$ is the number of ECG words in the vocabulary and $D$ is the dimension of each ECG word. Given an ECG signal sample $X \in \mathbb{R}^{C \times T}$, it is first encoded by the QRSTokenizer into an ECG sentence $x \in \mathbb{R}^{l \times t}$. To enhance the representation of the input data, we upscale the ECG sentence using ECGFormer backbone network described in Section 2.2 to obtain the ECG sentence embedding $e \in \mathbb{R}^{l \times D}$. For the interval representations $P = \{p_i | i = 1, \ldots, l\}$ in the ECG sentence embedding, we use a quantizer to convert them into ECG word embeddings. The ECG vocabulary looks up the nearest neighbor of each interval representation $p_i$ in $V$. We use cosine similarity to find the closest ECG word embedding. This procedure can be formulated as

$$z_i = \arg \min \|\ell_2(p_i) - \ell_2(v_i)\|_2$$

where $v_i$ is the quantized ECG word embedding, and $\ell_2$ represents $\ell_2$ normalization.

*2.3.2 Heartbeat Reconstruction.* Due to the high signal-to-noise ratio of ECG signals, directly reconstructing the original signal can train a highly efficient ECG vocabulary. After being labeled by the quantizer, the normalized discrete ECG word embeddings $\{\ell_2(z_i) | i = 1, \ldots, l\}$ are fed into the transformer decoder, which contains multiple Transformer blocks. This process can be represented as

$$\hat{x} = \bigcup_{i=1}^{l} f_d\left(\ell_2(v_{z_i})\right)$$

where $\hat{x}$ is the reconstructed ECG sentence and $f_d$ is the decoder.

To make the update of the ECG vocabulary more stable, we adopt an exponential moving average (EMA) strategy. The mean squared error (MSE) loss is utilized to guide the reconstruction. Finally, the loss function for training the vector-quantized heartbeat

**Table 1: Linear probing results of HeartLang and other methods. The best results are bolded, with  gray  indicating the second highest.**

| Method | PTBXL-Super | | | PTBXL-Sub | | | PTBXL-Form | | | PTBXL-Rhythm | | |
|---|---|---|---|---|---|---|---|---|---|---|---|---|
| | 1% | 10% | 100% | 1% | 10% | 100% | 1% | 10% | 100% | 1% | 10% | 100% |
| Random Init-Small | 54.50 | 69.59 | 80.75 | 53.28 | 62.08 | 75.51 | 53.46 | 54.32 | 61.46 | 61.06 | 68.42 | 76.95 |
| Random Init-Base | 61.26 | 73.23 | 81.76 | 55.11 | 63.90 | 77.61 | 47.36 | 50.03 | 61.94 | 49.19 | 57.23 | 69.99 |
| Random Init-Large | 67.04 | 78.87 | 83.51 | 55.63 | 65.69 | 77.86 | 50.54 | 53.38 | 63.95 | 58.43 | 67.65 | 80.79 |
| SimCLR[2] | 63.41 | 69.77 | 73.53 | 60.84 | 68.27 | 73.39 | 54.98 | 56.97 | 62.52 | 51.41 | 69.44 | 77.73 |
| BYOL[7] | 71.7 | 73.83 | 76.45 | 57.16 | 67.44 | 71.64 | 48.73 | 61.63 | 70.82 | 41.99 | 74.40 | 77.17 |
| BarlowTwins[18] | 72.87 | 75.96 | 78.41 | 62.57 | 70.84 | 74.34 | 52.12 | 60.39 | 66.14 | 50.12 | 73.54 | 77.62 |
| MoCo-v3[4] | 73.19 | 76.65 | 78.26 | 55.88 | 69.21 | 76.69 | 50.32 | 63.71 | 71.31 | 51.38 | 71.66 | 74.33 |
| SimSiam[3] | 73.15 | 72.7 | 75.63 | 62.52 | 69.31 | 76.38 | 55.16 | 62.91 | 71.31 | 49.3 | 69.47 | 75.92 |
| TS-TCC[6] | 70.73 | 75.88 | 78.91 | 53.54 | 66.98 | 77.87 | 48.04 | 61.79 | 71.18 | 43.34 | 69.48 | 78.23 |
| CLOCS[10] | 68.94 | 73.36 | 76.31 | 57.94 | 72.55 | 76.24 | 51.97 | 57.96 | 72.65 | 47.19 | 71.88 | 76.31 |
| ASTCL[17] | 72.51 | 77.31 | 81.02 | 61.86 | 68.77 | 76.51 | 44.14 | 60.93 | 66.99 | 52.38 | 71.98 | 76.05 |
| CRT[19] | 69.68 | 78.24 | 77.24 | 61.98 | 70.82 | 78.67 | 46.41 | 59.49 | 68.73 | 47.44 | 73.52 | 74.41 |
| ST-MEM[14] | 61.12 | 66.87 | 71.36 | 54.12 | 57.86 | 63.59 | 55.71 | 59.99 | 66.07 | 51.12 | 65.44 | 74.85 |
| MERL[11] | **82.39** | **86.27** | **88.67** | **64.9** | **80.56** | 84.72 | 58.26 | **72.43** | **79.65** | 53.33 | 82.88 | 88.34 |
| HeartLang-Small(Ours) | 70.61 | 84.23 | 87.89 | 58.08 | 73.68 | **87.12** | **59.61** | 65.28 | 77.61 | **69.95** | **83.26** | **92.36** |
| HeartLang-Base(Ours) | 71.44 | 83.97 | 87.96 | 60.55 | 76.60 | 86.66 | 57.76 | 63.14 | 77.43 | 56.32 | 71.71 | 90.54 |
| HeartLang-Large(Ours) | 69.05 | 83.65 | 87.02 | 57.95 | 72.95 | 85.59 | 56.44 | 61.15 | 73.33 | 54.41 | 72.38 | 90.15 |

reconstruction process is defined as

$$\mathcal{L}_V = \sum_{x \in \mathcal{D}} \sum_{i=1}^{l} \left( \|\hat{x}_i - x\|_2^2 + \|sg(\ell_2(p_i)) - \ell_2(v_{z_i})\|_2^2 + \|\ell_2(p_i) - sg(\ell_2(v_{z_i}))\|_2^2 \right)$$

where $D$ represents all the ECG sample data, and $sg$ denotes the stop-gradient operation, which is defined as the identity function in the forward pass and has zero gradient.

## 2.4 Masked Pre-Training

In this section, we introduce the pre-training part of the Heart-Lang framework, as depicted in the lower half of figure 2. This is a focal point of the entire HeartLang framework, aimed at utilizing unmasked parts of heartbeats and ECG Vocabulary containing heartbeat semantic information within ECG sentences. This forces the ECGFormer to learn rhythm-level features of ECG sentences, which inherently possess generalizability, thus effectively applying the rhythm-level knowledge learned by the ECGFormer to various downstream tasks. The pre-training component primarily includes ECG heartbeat masking and ECG sentence masking prediction.

*2.4.1 ECG Heartbeats Mask.* To enable the ECGFormer to learn the rhythm-level features of ECG sentences, we perform random masking on the entire ECG heartbeat sentence, allowing the ECG-Former to understand the content of the entire ECG sentence based on the unmasked heartbeats. For the heartbeat sentence $X \in \mathbb{R}^{l \times D}$ obtained through the QRS Tokenizer, where each vector can be

represented as:

$$e = \{e_i | i = 1, \ldots, l\}$$

We randomly generate a mask $M = \{m_i | i = 1, \ldots, l\}$ where $m_i \in \{0, 1\}$, setting the masking rate to 0.5, meaning that half of the heartbeats in each ECG heartbeat sentence are masked. We use a learnable mask token $e_M \in \mathbb{R}^d$ to replace the masked heartbeats. Thus, the entire heartbeat sentence can be represented as:

$$X = \{e_i : m_i = 0 | i = 1, \ldots, N\} \cup \{e_M : m_i = 1 | i = 1, \ldots, N\}$$

It is then enhanced with temporal and spatial position encoding, before being input to the Transformer's Encoder.

*2.4.2 Predict Vocabularies.* ECG sentences are first vector quantized as described in section 2.4.1, obtaining the indices of each word in the ECG sentence. Subsequently, we extract the indices of the words corresponding to the masked parts of the ECG sentence to achieve a one-to-one correspondence between ECG words and masks. The task of this stage is to predict the word indices of the masked parts based on the unmasked parts of the ECG sentence, by minimizing the discrepancy between the predicted word indices and the true word indices, thus facilitating model training. Therefore, the pre-training part uses the information in the ECG sentence itself as labels, with the ECG sentence as the input, representing a typical form of self-supervised learning. The target loss function

**Table 2: Different fine-tuning strategies results of HeartLang. The best results are bolded, with** gray **indicating the second highest.**

| Dataset | Trainable Layer | Parameter Size | | |
|---|---|---|---|---|
| | | Small | Base | Large |
| PTBXL-Super | Linear | 87.89 | 87.96 | 87.02 |
| | All | 90.37 | **90.90** | 90.36 |
| PTBXL-Sub | Linear | 87.12 | 86.66 | 85.59 |
| | All | **90.05** | 89.79 | 87.56 |
| PTBXL-Form | Linear | 77.61 | 77.43 | 73.33 |
| | All | **82.36** | 76.49 | 81.92 |
| PTBXL-Rhythm | Linear | **92.36** | 90.54 | 90.15 |
| | All | 92.31 | 91.10 | 91.02 |

for this stage is:

$$\mathcal{L}_P = -\sum_{x \in \mathcal{D}} \sum_{m_i=1} \log p(v_i|X)$$

## 3 EXPERIMENTS AND RESULTS

### 3.1 Dataset

The PTBXL dataset is utilized for pre-training and fine-tuning downstream tasks. This dataset comprises 21,837 ECG signals collected from 18,885 patients. Each sample consists of a 12-lead ECG recorded for 10 seconds at a sampling rate of 500 Hz. The dataset is annotated according to SCP-ECG and includes four multi-label classification subsets: Diagnostic superclass (5 classes), Diagnostic subclass (23 classes), Form (19 classes), and Rhythm (12 classes). Each downstream task category contains a varying number of samples. The official data split is followed for training, validation, and testing.

### 3.2 Implementation

All samples in the PTBXL dataset are first downsampled to 100 Hz to enhance computational efficiency. The construction of the ECG vocabulary and the pre-training of HeartLang are performed using all samples from the PTBXL dataset. Notably, no labels are required for training in these two tasks. We propose three variants of HeartLang with different parameter sizes: HeartLang-Small, HeartLang-Base, and HeartLang-Large. The specific parameter configurations for each model are detailed in the appendix A. In downstream fine-tuning tasks, two evaluation methods are proposed: full-parameter fine-tuning and linear probing. For full-parameter fine-tuning, none of the layers in ECGFormer are frozen, allowing all layers to be trained. For linear probing, the ECGFormer encoder is kept frozen, and only the parameters of the newly initialized linear classifier are updated. Each task is tested using 1%, 10%, and 100% of the training data to evaluate the generalization capability of HeartLang in few-shot scenarios. For all downstream tasks, macro AUC is used as the evaluation metric.

### 3.3 Evaluation on Linear Probing

We first evaluate our method using linear probing, as presented in Table 1. It is noteworthy that the comparative methods' results mainly come from MERL, one of whose major contributions is providing a benchmark from many eSSL methods. It must be stated that the results in MERL were pre-trained on the MIMIC-IV dataset, while in our work, due to computational resource limitations, the HeartLang were only pre-trained on PTB-XL. The sample size of MIMIC-IV is about thirty times larger than that of PTB-XL. Finally, MERL is not an eSSL method but a multimodal method incorporating text, whereas the other comparative methods are all eSSL methods.

The results show that our method demonstrates strong competitiveness in linear probing. Specifically, in the four downstream tasks, HeartLang-Base and HeartLang-Small achieved first or second highest in most cases with varying proportions of training data. Particularly, compared to other eSSL methods, our method almost universally outperformed them, except slightly lower performance in the 1% training sample cases of Diagnostic superclass and Diagnostic subclass classification tasks. Compared to MERL, our method achieved comparable or better results in a single modality (i.e., ECG signals alone). Compared to Random Init, where the encoder weights are randomly initialized, our method shows significant improvement. This indicates that our method indeed learns multi-level ECG representations, resulting in competitive performance in downstream tasks. Interestingly, we found that as the parameter size increased, HeartLang-Large did not bring a significant performance boost, which we suspect is due to the limited sample size of our pre-training dataset.

### 3.4 Evaluation on different fine-tuning strategies

Table 2 presents the performance results of different fine-tuning strategies. "Linear" represents linear probing, which only trains the final linear classification head while freezing the encoder. "All" represents full parameter fine-tuning, where all layers of the model are trained. The results show that, in most cases across the four downstream datasets, the full parameter fine-tuning strategy outperforms the linear probing strategy. Only in rare cases, such as HeartLang-Small on PTBXL-Rhythm and HeartLang-Base on PTBXL-Form, does the performance slightly fall behind the linear probing strategy. Additionally, most of the best results are concentrated in the HeartLang-Small model. We speculate that this is because the downstream tasks have relatively small amounts of data, and using models with very large number of parameters may lead to overfitting, thus reducing performance.

### 3.5 Case Study

We selected 256 samples for visual analysis of the ECG vocabulary, with specific results shown in Appendix B. To more prominently display the original forms of ECG words in different ECG vocabularies, we visualized the top 20 most frequently occurring ECG words, with the results shown in Figure 3. The distinct clusters observed in the visualization suggest that there is a natural grouping of ECG words based on their semantic or morphological similarities. Furthermore, the proximity of points to one another can provide

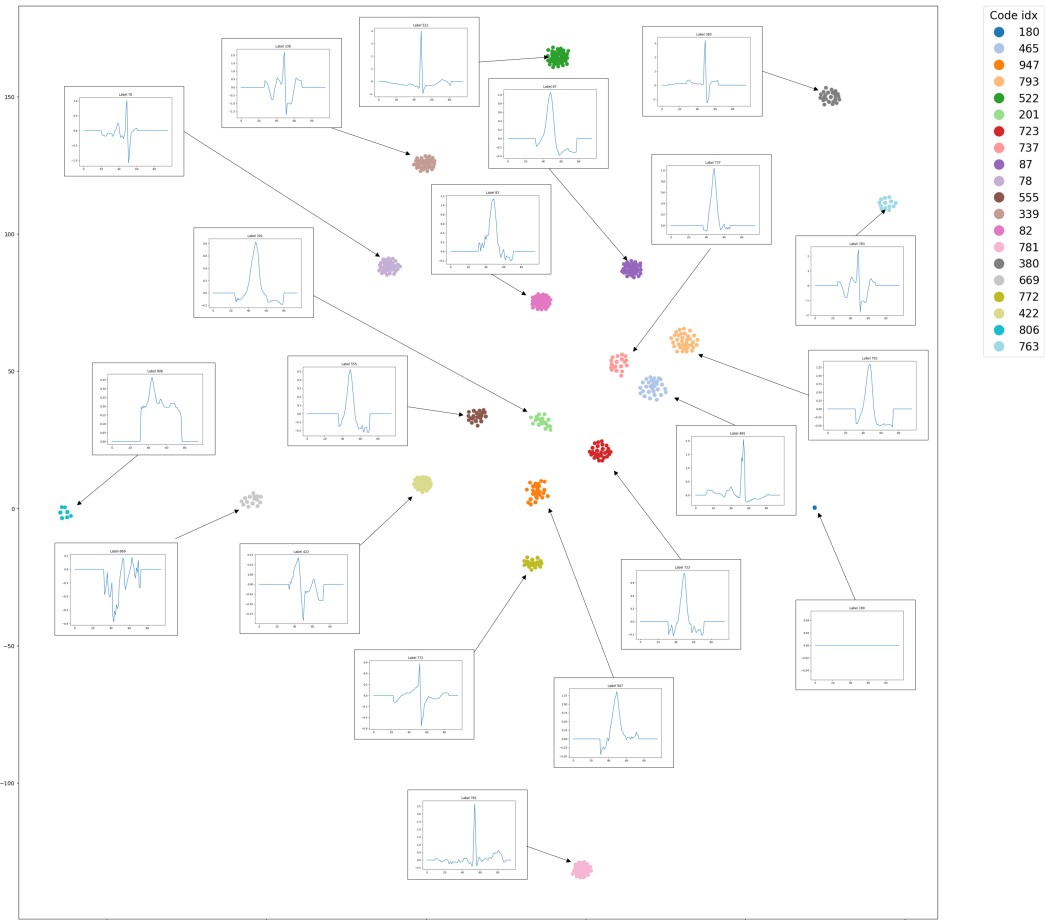

**Figure 3: Top 20 Most Frequently Occurring ECG Words.**

insights into the relationships between different ECG words. ECG words that are closely located may share similar characteristics and may jointly influence the occurrence of different diseases.

## 4  CONCLUSION AND FUTURE WORK

In this paper, we introduce HeartLang, a novel self-supervised framework for ECG language processing. This framework learns heartbeat-level and rhythm-level representations by constructing an ECG vocabulary and pre-training by masked prediction on ECG sentences, uncovering the latent semantic relationships in ECG signals. We provide a novel perspective on ECG signals by viewing heartbeats as words and rhythms as sentences, and use this approach to conduct large-scale pre-training on a vast amount of unlabeled ECG signals. Extensive experiments are conducted on benchmark datasets with varying proportions of training data, evaluating the performance of HeartLang under three parameter scales. The results show that our method exhibits significant advantages over other comparative eSSL methods across different scenarios.

In the future, we will further explore the effectiveness of the pre-training framework on larger-scale datasets. We hope our work provides an effective solution for future ECG foundational models and benefits the entire ECG representation learning research community.

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

## A  HYPERPARAMETERS SETTINGS

We have detailed the hyperparameters of the three different parameter scales of the HeartLang framework proposed in this paper, as well as the hyperparameters of various training methods used during training, in Table 3. In this paper, the hyperparameters used for training the three parameter scales of the framework are consistent.

## B  VISUALIZATION

We visualized the results of Vector-Quantized Heartbeat Reconstruction under the HeartLang-Base framework, as shown in Figure 4. The convergence curves of the total pre-training loss and masked modeling accuracy for the three configurations of HeartLang are shown in Figure 5. We have visualized the positional relationships of most ECG Words in the ECG Vocabulary by selecting 256 samples, as shown in Figure 6. It is worth noting that the outlier within the hollow circle is an ECG word that has undergone zero-padding.

Received 20 February 2007; revised 12 March 2009; accepted 5 June 2009

**Table 3: Hyperparameters for masked Pre-Training**

| Hyperparameters | HeartLang-Small | HeartLang-Base | HeartLang-Large |
|---|:---:|:---:|:---:|
| Transformer encoder layers | 6 | 12 | 24 |
| Hidden size | 768 | 768 | 768 |
| MLP size | 2048 | 2048 | 2048 |
| Attention head number | 4 | 10 | 16 |
| Batch size | | 256 | |
| Peak learning rate | | 5e-4 | |
| Minimal learning rate | | 1e-5 | |
| Learning rate scheduler | | Cosine | |
| Optimizer | | AdamW | |
| Adam $\beta$ | | (0.9,0.98) | |
| Weight decay | | 0.05 | |
| Total epochs | | 100 | |
| Warmup epochs | | 20 | |
| Gradient clipping | | 3.0 | |
| Mask ratio | | 0.5 | |

Jiarui Jin, Haoyu Wang, Jun Li, Sichao Huang, Jiahui Pan, and Shenda Hong

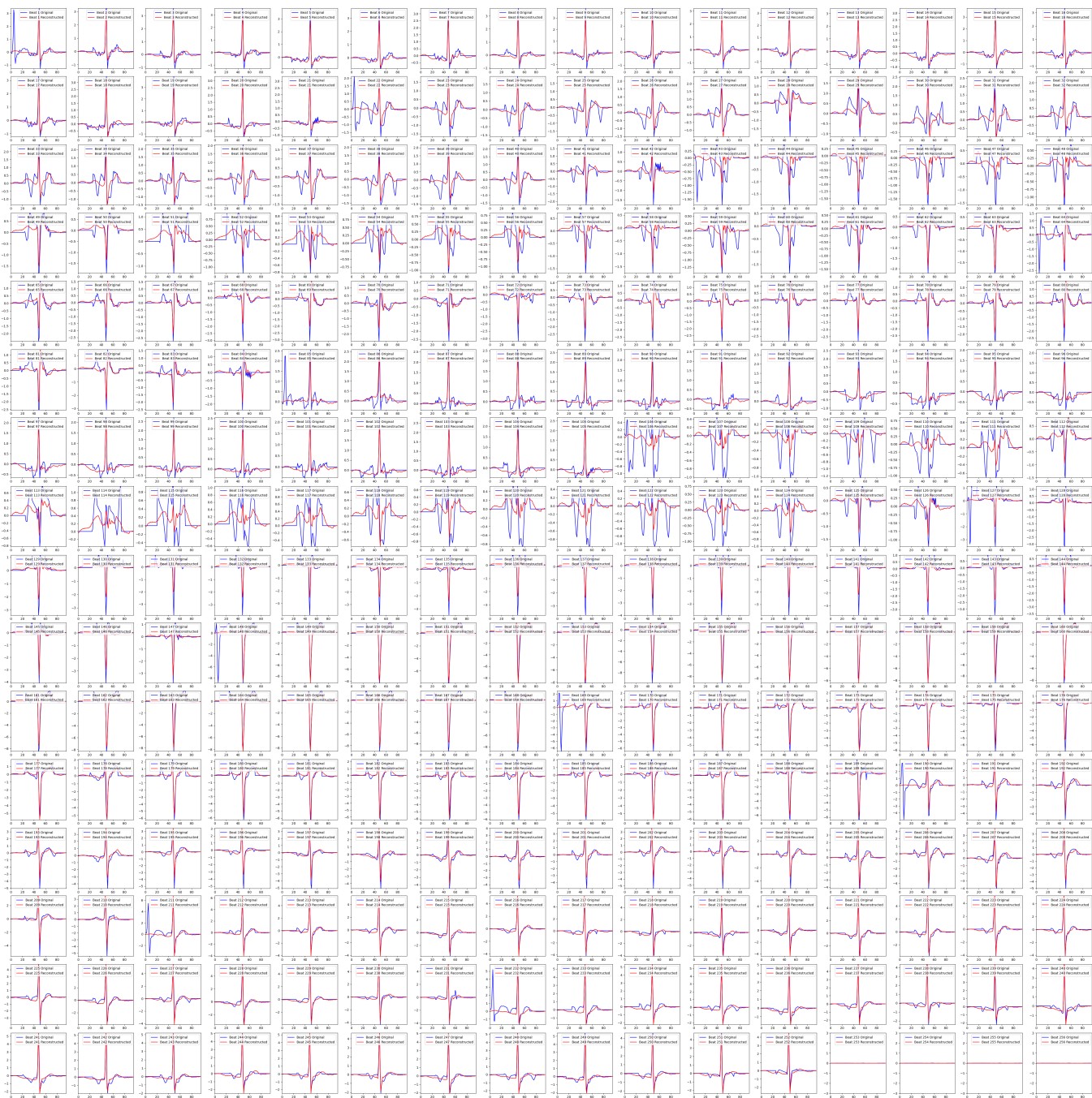

**Figure 4: Vector-Quantized Heartbeat Reconstruction Results.**

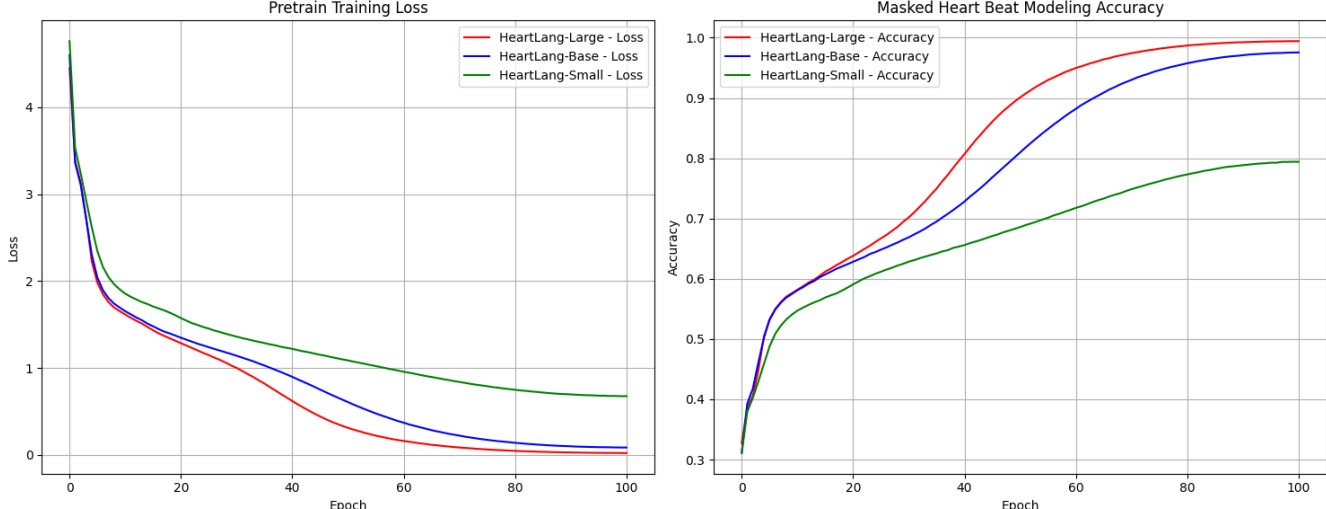

**Figure 5: Masked Pre-training Results.**

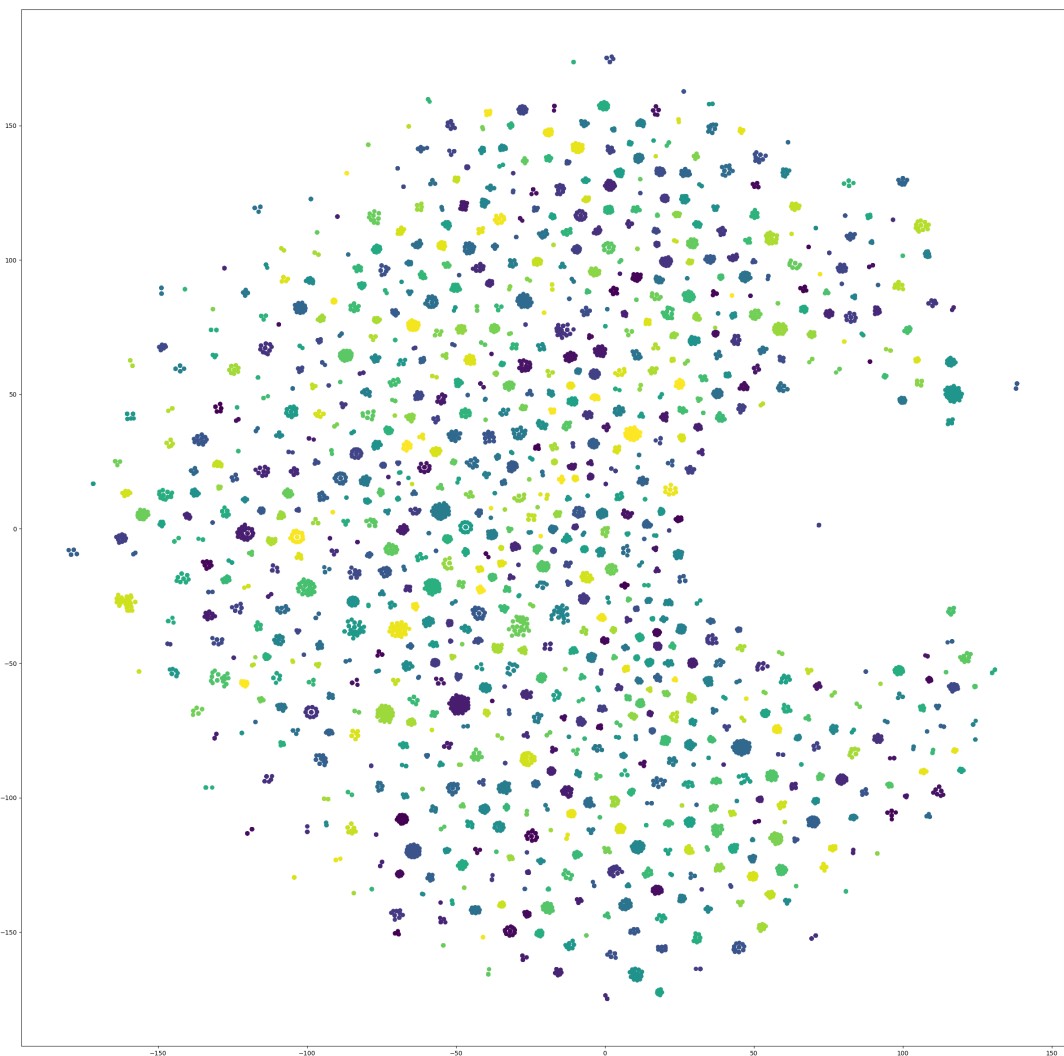

**Figure 6: ECG Vocabulary Visualization.**