# OpenReview forum: "Reading Your Heart: Learning ECG Words and Sentences via Pre-training ECG Language Model"
_KDD.org/2024/Workshop/AIDSH — KDD-AIDSH 2024 Oral_

### Official Review · Reviewer_fdyr · 2024-06-11
**Reading Your Heart: Learning ECG Words and Sentences via Pre-training ECG Language Model**

**Rating:** 8
**Confidence:** 3

**Review:**

Pre-training ECG as an NLP task is a very interesting approach. By converting complex time series pre-training methods to well-explored NLP pre-training algorithms, researchers are often able to gain more insights. Some well-tested NLP algorithms can also be applied to the ECG domain, such as BERT. I believe that this work, as well as subsequent efforts, will contribute to the community. At the same time, we are eager to see the creation of a more comprehensive vocabulary and the resolution of heartbeat signal encoding in some extreme cases in upcoming works.

Strength:

- Using a tokenizer can effectively reduce the computational load during the pre-training process. Compared to directly computing time series, computing sentences is more efficient.
- The author converts complex ECG signals into words and sentences, and trains the ECG signals using a BERT-like pre-training method. This approach integrates knowledge from more mature fields into the realm of ECG signal processing.

Weakness:

- The QRSTokenizer used in this paper seems to have lost some key features when encoding ECG waveforms, such as heart rate. Many conditions of heart diseases are characterized by abnormal waveforms under high heart rate states. After Tokenization, the length of a sentence to some extent represents its heart rate, but truncation will lead to the loss of crucial information.

Question:
- Unlike NLP, ECG signals do not have clear BOS and EOS; there is inevitably some information loss at the beginning and end of the signals. For example, if a signal is missing a Q wave due to collection issues, how should these incomplete 'words' be handled?
- Some abnormal heartbeats do not have clear QRS waves, such as ventricular beat and cardiac arrest. Even some ECG equipment may pick up noise. Can the ECG vocabulary cover these situations?
- In Section 2.3, Figure X is missing. Also, I do not quite understand the purpose of Section 2.3; why is it necessary to use reconstruction loss for trying word embeddings? Is this loss one of the optimization targets during the pre-training process, or is this loss function only used for creating the vocabulary?

---

### Official Review · Reviewer_PKjH · 2024-06-19

**Rating:** 6
**Confidence:** 3

**Review:**

Summary:
This paper introduces a novel self-supervised learning framework called HeartLang for ECG language processing. It highlights the understanding of ECG signals by treating heartbeats as words and rhythms as sentences. By constructing an ECG vocabulary and taking advantage of masked prediction during pre-training, HeartLang learns both heartbeat-level and rhythm-level representations, uncovering latent semantic relationships in ECG signals. The framework offers three scales (HeartLang-Small, HeartLang-Base, and HeartLang-Large) which outperforms other eSSL methods in experiments. This innovative approach paves the way for more effective ECG foundational models and benefits the ECG representation learning field.

Strengths:
1. The paper is well-written and organized.
2. The paper introduces an innovative perspective on ECG signals by viewing heartbeats as words and rhythms as sentences, and
extract latent semantic representations in ECG signals using self-supervised learning.
3. The proposed framework is simple yet effective, which is shown in the experiments.

Weaknesses:
1. The datasets used in the paper is not large enough for evaluating the scalability.
2. The method proposed enables training of all layers, but what about its efficiency? The experiments include baseline models that only fine-tune linear layers. Is Lora utilized in this fine-tuning process? The experimental section lacks detailed information on the techniques employed.

---

### Decision · Program_Chairs · 2024-06-28

Accept (Oral)